# A Review of MicroRNAs and lncRNAs in Atherosclerosis as Well as Some Major Inflammatory Conditions Affecting Atherosclerosis

**DOI:** 10.3390/biomedicines12061322

**Published:** 2024-06-13

**Authors:** Jernej Letonja, Danijel Petrovič

**Affiliations:** 1Institute of Histology and Embryology, Faculty of Medicine, University of Ljubljana, Korytkova 2, 1000 Ljubljana, Slovenia; jernej.letonja@mf.uni-lj.si; 2Laboratory for Histology and Genetics of Atherosclerosis and Microvascular Diseases, Institute of Histology and Embryology, Faculty of Medicine, University of Ljubljana, Korytkova 2, 1000 Ljubljana, Slovenia

**Keywords:** atherosclerosis, psoriasis, type 2 diabetes mellitus, rheumatoid arthritis, microRNA, extracellular vesicles, long non-coding RNA

## Abstract

It is generally accepted that atherosclerosis is a chronic inflammatory disease. The link between atherosclerosis and other inflammatory diseases such as psoriasis, type 2 diabetes mellitus (T2DM), and rheumatoid arthritis (RA) via metabolic, inflammatory, and immunoregulatory pathways is well established. The aim of our review was to summarize the associations between selected microRNAs (miRs) and long non-coding RNAs (lncRNAs) and atherosclerosis, psoriasis, T2DM, and RA. We reviewed the role of miR-146a, miR-210, miR-143, miR-223, miR-126, miR-21, miR-155, miR-145, miR-200, miR-133, miR-135, miR-221, miR-424, let-7, lncRNA-H19, lncRNA-MEG3, lncRNA-UCA1, and lncRNA-XIST in atherosclerosis and psoriasis, T2DM, and RA. Extracellular vesicles (EVs) are a method of intracellular signal transduction. Their function depends on surface expression, cargo, and the cell from which they originate. The majority of the studies that investigated lncRNAs and some miRs had relatively small sample sizes, which limits the generalizability of their findings and indicates the need for more research. Based on the studies reviewed, miR-146a, miR-155, miR-145, miR-200, miR-133, and lncRNA-H19 are the most promising potential biomarkers and, possibly, therapeutic targets for atherosclerosis as well as T2DM, RA, and psoriasis.

## 1. Introduction

Cardiovascular disease (CVD) is the leading cause of death globally. The estimated prevalence was 523 million, and CVD was the cause of death for an estimated 18.6 million people in 2019 [1]. It is now widely accepted that inflammation plays an important role in atherosclerosis and cardiovascular disease (CVD). It is also involved in all phases of the atherosclerotic process, from early endothelial dysfunction to advanced atherosclerotic plaques [2,3]. People with inflammatory diseases such as psoriasis, rheumatoid arthritis, chronic obstructive pulmonary disease, diabetes, and periodontitis may be at increased risk for CVD [2,3].

Elevated circulating markers such as C-reactive protein (CRP), interleukins (IL) (IL-1β, IL-6, IL-8, IL-12, IL-17, IL-32, IL-36), tumor necrosis factor α (TNFα), and vascular endothelial growth factor (VEGF) are commonly found in conditions of low-grade inflammation, such as CVD, psoriasis, RA, and diabetes mellitus (DM) [4,5,6]. Immune cells of innate or adaptive immunity (monocytes, neutrophils, mast cells, macrophages, and lymphocytes) are thought to play an important role in chronic inflammation [6,7]. Moreover, they were reported to be responsible for the production of inflammatory cytokines [4,5,6].

Psoriasis is a multifactorial disease with a strong genetic background [8,9,10]. Psoriasis affects approximately 2.5% of the population worldwide [8,11]. HLA-Cw6 is one of the most strongly associated psoriasis susceptibility alleles [8]. Patients with psoriasis can also be affected by psoriatic arthritis and are at greater risk for CVD and metabolic syndrome [12,13]. The risk of developing CVD or metabolic syndrome is dependent on the severity of psoriasis [14]. A large cohort study reported a 57% greater risk for CVD in patients with severe psoriasis after adjusting for major traditional risk factors.

Psoriasis is considered to primarily be a T cell-mediated immune disease, and Th1 and Th17 lymphocytes are expected to be the main players via the release of inflammatory cytokines (i.e., IL-1α, IL-1β, IL-17, IL-22, TNF-α), leading to keratinocyte proliferation, the migration of inflammatory cells, and enhanced inflammatory response in the skin [15,16,17]. TNF-α and IL-1 are the driving forces of chronic systemic inflammation predisposing psoriasis patients to CVD and metabolic syndrome [15].

Type 2 diabetes mellitus (T2DM) is a chronic multisystemic disease that is characterized by hyperglycemia and insulin resistance. The prevalence is increasing; 784 million people are expected to suffer from T2DM by 2045 [18]. Metabolic syndrome (high blood pressure, hyperglycemia, abdominal obesity, high serum triglycerides, and low serum high-density cholesterol (HDL)) is a major predisposing factor for both CVD and T2DM. CVD is a major cause of death for patients with T2DM; a study reported that CVD was the cause of death in 50.3% of patients with T2DM [6,19]. Hyperglycemia promotes the formation of reactive oxygen species (ROS) that are involved in endothelial dysfunction and inflammation [6].

Rheumatoid arthritis (RA) is a chronic inflammatory disease that mainly affects the joints; however, extra-articular involvement is not uncommon [20]. The global prevalence of RA in 2017 was 0.27%; however, there is a strong geographical and socioeconomic factor to the disease (the prevalence is higher in northern latitudes and higher-income countries) [21]. Dendritic cells play a crucial role in initiating and maintaining inflammation in RA. They modulate Th1, Th2, and Th17 lymphocyte activity by secreting inflammatory cytokines (IL-1 β, IL-6, IL-12, IL-18, IL-23, TGF-β). TNF-α, IL-1 β, and IL-6 play a key role in establishing and maintaining inflammation in the synovia [5].

MicroRNAs (miRs) are classified as short, non-coding RNA molecules that are involved in cell signaling, intracellular communication, and gene expression [22]. They regulate physiological and pathological processes, including autoimmune disorders, metabolism, inflammation, and cancerogenesis [23]. In the last couple of years, research into microRNAs has yielded several specific microRNAs that have been identified as being involved in atherosclerosis and other inflammatory diseases.

Long non-coding RNA (lncRNA) are RNA transcripts that are longer than 200 nucleotides and do not encode proteins [24]. In recent years, they have been extensively studied, and it has been discovered that they are involved in various cellular processes such as apoptosis, metabolism, inflammation, cell differentiation, and proliferation [24].

Extracellular vesicles (EV) are a method of intercellular signaling used by almost all cells in the human body. Extracellular vesicles consist of a bilipid layer membrane that encapsulates a cargo of proteins, lipids, mRNA, and microRNA [25]. Their classification is still controversial, but they can be divided into three categories based on their size: small or exosomes (<100 nm in diameter), medium or microvesicles (MV) (100–1000 nm in diameter), and large or apoptotic bodies (>1000 nm in diameter) [26]. The function of EVs is highly dependent on their surface expression as well as their content [27].

The aim of our review was to summarize the associations between selected microRNAs (miRs) and long non-coding RNAs (lncRNAs) and atherosclerosis, psoriasis, T2DM, and RA.

## 2. Methods

We reviewed articles that investigated the roles of genetic and epigenetic factors in the pathogenesis of atherosclerosis and psoriasis, T2DM, and RA. We used the PubMed database as our main bibliographic source. The following search sequence was used to obtain articles: (atherosclerosis) AND ((psoriasis)) OR (T2DM) OR (diabetes) OR (type 2 diabetes mellitus) OR (rheumatoid arthritis) OR (RA)) AND ((microRNA) OR (miR) OR (long non-coding RNA) OR (lncRNA) OR (extracellular vesicles) OR (EVs) OR (exosomes) OR (microvesicles) OR (apoptotic bodies)).

Only articles written in English were included. Abstracts of all acquired articles with relevant titles were read. Articles that had promising abstracts were then read in their entirety. A targeted search of the PubMed database with specific search terms (e.g., (psoriasis) AND/OR (atherosclerosis) AND ((miR-200) OR (microRNA-200))) was then conducted to search for potential additional articles. We also checked publications that were cited in the articles that we found. The last search in the PubMed database was conducted in January 2024. Figure 1 schematically represents our article selection process.

## 3. Pathogenesis of Atherosclerosis in Selected Inflammatory Conditions

Several studies have shown that psoriasis and atherosclerosis are linked by dyslipidemia, increased levels of lipoprotein a, and altered metabolic, inflammatory, and immunoregulatory pathways [28,29,30]. In addition, the immune system is thought to play a crucial role in the pathogenesis of psoriasis, and various innate and adaptive immune cells, as well as proinflammatory mediators, are also involved in the development of atherosclerosis [2,3,28].

One of the crucial cytokines involved in the pathogenesis of psoriasis and RA, IL-17, has been reported to play an important role in the production of ROS [11,31]. ROS are on the list of factors that can cause endothelial dysfunction [7]. IL-17 is one of the main factors for endothelial dysfunction in patients with RA [32]. Hyperglycemia causes endothelial dysfunction in T2DM through increased oxidative stress via ROS and reactive nitrogen species (RNS) [33]. Endothelial dysfunction causes changes in the production of endothelial cells (i.e., expression of adhesion molecules, ROMO1 expression, and production of inflammatory cytokines …), and they lead to morphological changes in the vessel wall (inflammation, atherosclerotic changes) [2,3,7]. IL-1β, IL-6, IL-17, TNF-α, and hs-CRP are involved in the pathogenesis of atherosclerosis, psoriasis, and RA [34,35].

Endothelial dysfunction, measured by functional ultrasound studies of the brachial artery, and carotid intima-media thickness (CIMT) were reported to be important markers of subclinical atherosclerosis in psoriasis [36,37]. A meta-analysis involving psoriasis patients showed that psoriasis patients had a significantly thicker CIMT (WMD 0.11 mm; 95% CI 0.08–0.15) and impaired flow-mediated dilation (FMD) (WMD −2.79%; −4.14% to −1.43%) in comparison with the control group [37]. Psoriasis patients with a mean age > 45 years had a much thicker CIMT, while impaired FMD appeared to be more pronounced in psoriasis patients with a mean age of <45 years. The authors concluded in their meta-analysis that patients with psoriasis had an excessive risk of subclinical atherosclerosis [37].

Patients with T2DM have a significantly higher risk of cardiovascular events compared to people without T2DM [38]. A meta-analysis showed that the CIMT of patients with T2DM was thicker (0.13 (95% CI 0.12–0.14) mm thicker) compared to controls [39].

A meta-analysis of studies examining the CIMT in patients with RA showed that patients with RA have a significantly thicker CIMT compared to controls [40].

Dyslipidemia plays a key role in atherogenesis. Although the mechanisms by which low-density lipoprotein (LDL) provokes lesion formation are not fully understood, it is speculated that the oxidative modification of LDL particles leads to the accumulation of oxidized LDL in the vessel wall and the development and progression of plaques [2,3]. Oxidized LDL cholesterol (ox-LDL) has been reported to induce proinflammatory and proatherogenic effects via various mechanisms (e.g., via NF-κB, caspase-1 activation) [2,3,41,42]. Patients with psoriasis have an atherogenic lipid profile with elevated LDL cholesterol levels [43]. In addition, impaired HDL function in psoriasis patients also increases the production of oxidized LDL cholesterol [43].

Patients with T2DM have a characteristic proatherogenic lipid profile with low HDL levels and elevated LDL and triglyceride levels [44]. The so-called lipid paradox has been reported in patients with RA. In patients in the active stage of the disease, total cholesterol, LDL, and HDL levels are lowered, while they increase with the administration of drugs targeting the inflammatory pathways in RA. The relationship between lipid profile and CVD risk is, therefore, U-shaped. The anti-inflammatory and antiatherogenic functions of HDL are impaired in patients with RA [45].

Macrophages filled with oxidized LDL cholesterol (i.e., foam cells) play a crucial role in the development of the atherosclerotic process [2,3]. Foam cells and other antigen-presenting cells (e.g., dendritic cells) can present oxidized LDL particles and heat shock proteins to the innate and adaptive immune system, leading to a chronic, low-grade inflammatory response [42].

In addition, hemodynamic forces, via endothelial dysfunction, represent an important local risk factor for atherogenesis [46]. Atherosclerotic lesions are mainly located in areas where laminar flow is disturbed and turbulent flow is present (i.e., bifurcations of arteries) [46].

In the last 10–15 years, the introduction of biological therapies, such as TNF-α, IL-12/23, and IL-17 inhibitors in psoriasis and IL-6, CD20, TNF-α, and CD80/86 inhibitors in RA, is expected to influence the rate of atherothrombotic complications in patients with psoriasis and RA [8,11,47].

## 4. Gene Expression in Psoriasis, T2DM, and RA-Promoting Atherosclerosis Development

The gene expression profiles of psoriasis (GSE30999) and atherosclerosis (GSE28829) were downloaded from the Gene Expression Omnibus (GEO) database, and the common differentially expressed genes (DEGs) of psoriasis and atherosclerosis were identified. A functional analysis of DEGs emphasizes the important role of chemokines and cytokines in these two diseases. In addition, the lipopolysaccharide-mediated signaling pathway is closely related to both. Finally, 16 important hub genes were identified using cytoHubba, including *LCP2*, *CD53*, *LYN*, *CSF2RB*, *C1QB*, *MMP9*, *PLEK*, *PTPRC*, *FYB*, *IL1RN*, *RAC2*, *CCL5*, *IRF8*, *BCL2A1*, *NCF2*, and *TLR2* [48]. The results of the study revealed some common pathogenetic mechanisms of psoriasis and atherosclerosis [48].

The IL-23R rs6682925T/C polymorphism and the inheritance of the *HLA*, *FUT2*, *UBE2L3*, and *SH2B3* gene variants increase the risk of major adverse cardiovascular events (MACE) in psoriasis [16,49]. The *IL-23A* polymorphism rs2066808 is associated with an increased risk of developing psoriasis and could also increase the genetic risk for premature CAD [50,51].

Eder and colleagues conducted a study on 411 patients with psoriasis and discovered that the HLA-C*06-02 and HLA-B*13-02 alleles are associated with a higher risk of atherosclerosis [52].

Mutations in *CARD14*, which is an NF-κB regulatory protein, have been described in psoriatic patients. CARD14 is also expressed in the cells of the aortic endothelium, which could result in atherosclerotic progression and increased systemic inflammation [53].

Gene expression profiles of atherosclerosis and T2DM were obtained from the GEO database to identify DEGs of atherosclerosis and T2DM. GSE28829 (atherosclerosis) and GSE20966 (T2DM) were used as test sets, and the results were then validated with GSE43292 (atherosclerosis) and GSE25724 (T2DM). Genes related to immune activation and cytokines were found to have important roles in the pathogenesis of these diseases. Three important hub genes were identified using cytoHubba after validation with GSE43292 and GSE25724 (*CD4*, *PLEK*, and *THY1*). Further validation using clinical samples identified CD4 and PLEK as the key genes in atherosclerosis and T2DM [54].

Gene expression profiles of atherosclerosis (GSE14905) and RA (GSE55235 and GSE55457) were obtained from the GEO database, and DEGs were identified. A total of 12 important hub genes were identified using cytoHubba: *CYBB*, *LAPTM5*, *CSF1R*, *HCK*, *ITGAM*, *CD86*, *C1QA*, *ITGB2*, *PTPRC*, *CTSS*, *LCP2*, and *CD53*. Expression levels of the following genes were then verified: *CSF1R*, *CD86*, *PTPRC*, and *CD53*. Genes related to phagocytosis, neutrophil activation, and leukocyte migration were identified as being significantly enriched in the pathogenesis of RA and atherosclerosis [55].

## 5. The Role of MicroRNAs in Atherosclerosis, Psoriasis, T2DM, and RA

### 5.1. MiR-146a

MiR-146a has an anti-inflammatory function as it inhibits the NF-κB pathway and modulates the expression of cytokines (IL-6, IL-8, IL-17, TNFα, and others) [56,57]. It also mediates the proliferation and migration of VSMCs [58,59]. Elevated levels of miR-146a have been found in atherosclerotic lesions and in the blood of patients with psoriasis. [56,60,61,62]. Leal and colleagues also reported a significant association between miR-146a levels and the PASI score, as well as the body surface area index [56]. Ele-Refaei and El-Esawy described a decrease in miR-146a levels after 12 weeks of aggressive treatment [62]. Vaher and colleagues researched miR-146a levels in skin samples obtained by a punch biopsy which indicates that the upregulation of miR-146a is both systemic as well as localized to the affected skin [60].

The expression levels of miR-146a correlate with the expression levels of IL-6 and TNFα in patients with atherosclerosis. MiR-146a levels are higher in patients with carotid atherosclerosis and correlate with the degree of stenosis and stability of the atherosclerotic plaque [57,63,64,65,66]. Zhelankin and colleagues, as well as Gao et al., investigated miR-146a levels in patients with coronary atherosclerosis [64,66]. Zhelankin et al. concluded that miR-146a-5p could be used as a biomarker for acute coronary syndrome (ACS) [64]. However, their study included 50 participants and 30 controls, so more studies with larger participant groups are needed. Huang et al. included 180 participants with carotid atherosclerosis and further divided them into groups based on the degree of stenosis and plaque stability [57]. The study by Guo et al. investigated the plasma levels of miR-146a in 100 newly diagnosed patients with T2DM and reported an association between the miR-146a levels and CIMT [65]. Raitoharju et al. reported increased miR-146a expression in femoral and aortic plaques but not in carotid plaques [63]. MiR-146a polymorphism rs2910164 has been extensively studied in the context of atherosclerosis and psoriasis. The G allele of rs2910164 is associated with an increased risk of psoriasis [67]. Two meta-analyses concluded that it may be associated with a lower risk of CHD, but further research is needed [68,69].

Alipoor et al. performed a meta-analysis and concluded that miR-146a expression is downregulated in whole blood and PBMCs from patients with T2DM compared to controls [70]. A more recent meta-analysis by Zhu and Leung also concluded that miR-146a is downregulated in PBMCs but also that it is upregulated in adipose tissue [71].

However, a meta-analysis concluded that the rs2910164 polymorphism is not associated with susceptibility to T2DM [72]. Plasma miR-146a is a marker of subclinical atherosclerosis in patients with T2DM and correlates with CIMT [65]. According to Shen and colleagues, the CC genotype of the rs2910164 polymorphism was associated with an increased risk of plaque vulnerability in patients with T2DM but not with carotid atherosclerosis [73].

A meta-analysis evaluated studies that researched miR-146a levels in patients with RA and concluded that circulating and PBMC levels of miR-146a are elevated and correlate with the erythrocyte sedimentation rate and disease activity in patients with RA [74]. The CC genotype of the rs2910164 polymorphism is considered a protective factor for RA in the Egyptian-Caucasian population, according to Liu et al. [75].

MiR-146a is an important biomarker in the inflammatory diseases studied in this article. It is upregulated in patients with atherosclerosis, psoriasis, and RA. It is downregulated in patients with T2DM but is recognized as a marker for atherosclerosis in patients with T2DM as it correlates with CIMT. MiR-146 could also be a useful marker for monitoring the effectiveness of the treatment of psoriasis, according to Ele-Refaei and El-Esawy [62].

The available data suggest that it is a promising biomarker for atherosclerosis in a healthy population as well as in patients with psoriasis, T2DM, or RA. The anti-inflammatory function of miR-146a has been reported in vitro and in animal studies [56]. Mir-146a modulates the Th1 response and directly inhibits parts of the NF-κB pathway. Because of this, the effects of miR-146a overexpression should be investigated as potential therapeutic options. New research is needed to investigate its potential as a biomarker for atherosclerosis in patients with psoriasis or RA.

### 5.2. MiR-210

MiR-210 is involved in VSMC migration and endothelial cell apoptosis [76]. Hypoxic conditions induce the expression of miR-210 in the endothelium, and its expression is also enhanced by TGF-β and IL-23, which is why it is extensively studied in cancer research [77,78].

The increased expression of miR-210 has been found both in the serum of patients with peripheral artery disease and in the atherosclerotic plaques themselves [63,79]. However, both studies were performed with a small sample size (≤30 per group). Eken and colleagues reported decreased miR-210 expression in unstable carotid plaques and concluded that miR-210 stabilizes the fibrotic cap of advanced atherosclerotic lesions [80].

MiR-210 also has an immunomodulatory function, affecting the levels of IL-10 and IL-17 and the differentiation of Th1 and Th17 cells [78,81]. Zhao et al. and Wu et al. reported increased miR-210 expression in CD4+ cells in patients with psoriasis compared to healthy controls [78,81]. Wu et al. also reported increased miR-210 expression in psoriatic plaques [78].

Plasma levels of miR-210 were elevated in patients with T2DM compared to healthy controls [82]. In another study, higher levels also correlated with higher BMI in T2DM patients [83]. Amr and colleagues reported increased plasma miR-210 levels in T2DM patients compared to healthy controls, as well as higher levels in T2DM patients with CHD compared to those without CHD [84]. Zhou and colleagues reported a decreased expression of miR-210 in erythrocytes compared to healthy controls [85]. A small study investigated the expression of miR-210 in plasma-derived extracellular vesicles in T2DM patients with ischemic heart disease. They reported a downregulation of miR-210 compared to healthy controls and concluded that plasma-derived extracellular vesicles and their cargo could be a potential biomarker for ischemic heart disease in T2DM [86]. Zhu and Leung concluded that miR-221 is significantly upregulated in the serum of T2DM patients and is one of the most important biomarkers for T2DM [71].

In patients with RA, serum levels of miR-210 were lower compared to healthy controls, and they were inversely correlated with TNF-α and IL-1β. Abdul-Maksoud et al. concluded that miR-210 could be used as a biomarker for RA [87]. However, Huang et al. did not report a statistically significant difference in PBMC miR-210 levels between patients with RA and healthy controls, but their sample size was much smaller [88].

The importance of miR-210 as a biomarker for T2DM and atherosclerosis in patients with T2DM makes it an interesting target for future studies and a potential target for treatments. Larger studies on patients with atherosclerosis, psoriasis, and RA are needed, as the sample sizes of the studies investigated were quite small (<50 patients per group). Mir-210 is intensively studied in cancer research because of its involvement in the response to hypoxic conditions and the immune response. The results from this research could be translated into potential treatments for atherosclerotic complications (e.g., myocardial infarction and ischemic stroke).

### 5.3. MiR-143

The expression of miR-143 in endothelial cells is upregulated by stable laminar blood flow [89]. It is then transferred to VSMCs via extracellular vesicles, where it has an antiatherogenic effect by regulating VSMC proliferation and differentiation [89,90]. MiR-143 has been shown to promote ROS production in mouse cardiomyocytes, but its role in oxidative stress in humans is not known [91].

In patients with ACS, plasma levels of miR-143 were significantly decreased compared to patients with coronary stenosis <50% and inversely correlated with the degree of coronary artery stenosis [92].

The role of miR-143 in psoriasis is still unclear. Løvendorf and colleagues reported significantly higher levels of miR-143 in PBMCs and a positive correlation with the PASI score [93]. A more recent study by Zheng and colleagues found decreased levels of miR-143 in patients with psoriasis vulgaris and a negative correlation with disease severity [94]. Their sample size was also larger; however, further studies are needed to evaluate the role of miR-143 in PBMC in psoriasis.

MiR-143-3p is significantly upregulated in VSMCs in patients with T2DM and is a promising biomarker for T2DM, according to the results of a meta-analysis conducted by Zhu and Leung [71]. Its overexpression has been associated with insulin resistance and the dysregulation of glucose metabolism in animal models.

It is also overexpressed in the plasma of patients with early RA with erosions of the second metacarpophalangeal bone in comparison to those without erosions. A positive correlation between its levels and CRP, as well as the clinical swollen joint count and subjective pain score, has also been described [95]. The study included 117 patients with RA and only 6 healthy controls because the main aim was to compare RA patients with erosions to those without. Studies with larger sample sizes with more healthy controls are needed.

MiR-143 is an interesting target for future studies of atherosclerosis in inflammatory diseases due to its effect on VSMCs and ROS.

### 5.4. MiR-223

MiR-223 is involved in the regulation of inflammation, cholesterol metabolism, and VSMC migration, proliferation, and apoptosis [96,97,98]. It is thought to have an anti-inflammatory function by suppressing the NLRP3 inflammasome, IL-1β, and IL-10 [98,99].

Singh and colleagues reported an upregulation of miR-223 in patients with unstable coronary artery disease [100]. They suggested that miR-223 may be a marker of plaque instability [100]. Guo and colleagues also found an increased expression of miR-223 in patients with coronary atherosclerosis and a significant correlation between miR-223 levels and disease severity [101]. On the other hand, Zhu and colleagues reported decreased miR-223 levels in patients with carotid atherosclerosis and a correlation with plaque stability, although their sample size was smaller [102]. Different atherosclerotic phenotypes (in this case, coronary and carotid atherosclerosis) could have different microRNA signatures, which could explain the opposing conclusions obtained by the authors of the above-mentioned studies. However, the studies by Singh et al. and Guo et al. included more participants than the study by Zhu et al. [100,101,102]. A study that would investigate the expression of miR-223 in carotid, coronary, and peripheral atherosclerosis on a large enough sample would be most beneficial.

MiR-223 promotes proliferation and inhibits apoptosis in keratinocytes [103]. Løvendorf and colleagues found increased levels of miR-223 in patients with psoriasis and a positive correlation with the PASI score [93]. MiR-223 was significantly downregulated after 3–5 weeks of treatment with methotrexate [93]. Pivarci and colleagues found no significant difference in miR-223 levels between patients with psoriasis and healthy controls but described a significant downregulation after treatment with etanercept [104]. However, Alatas and colleagues reported a significant downregulation of miR-223 in the blood of patients with psoriasis compared to healthy controls [105]. The correlation between miR-223 levels and psoriasis is unclear, as all three studies examined provided contradictory results with a comparable number of participants.

In the serum of patients with T2DM, miR-223 was significantly downregulated in circulating microvesicles and could even be used as a biomarker for progression from prediabetes to diabetes [99]. Zhu and Leung also concluded that miR-223 is significantly downregulated in the plasma of patients with T2DM [71].

Plasma levels of miR-223 were increased 2.5-fold in patients with RA compared to controls, but there was no correlation with disease activity [106]. We found only one study that investigated miR-223 in patients with RA.

MiR-223 is a potential biomarker of the stability of coronary plaques and the progression of prediabetes to diabetes.

### 5.5. MiR-155

The expression of miR-155 is stimulated by inflammation. TNF-α stimulates its transcription through NF-κB. Its transcription is also stimulated by ox-LDL. MiR-155 is thought to have an anti-inflammatory function as it suppresses TNF-α. In atherosclerosis, authors reported both an upregulation and downregulation of miR-155, though significant dysregulation remained consistent [107,108,109].

Fichtlscherer et al. reported decreased levels of miR-155 in the blood of patients with stable coronary heart disease (CHD) compared to healthy controls [107]. Wang and colleagues also reported reduced miR-155 levels in early coronary atherosclerotic plaques found in healthy heart donors [109]. However, a larger study by Li et al. found increased levels of miR-155 in plasma and atherosclerotic plaques [108]. The expression of miR-155 is stimulated by oxidized LDL and TNF-α [108]. Li and colleagues found that miR-155 has an anti-inflammatory function because it suppresses TNF-α expression [108].

MiR-155 is another microRNA whose levels correlate with the severity of psoriasis [110]. García-Rodríguez and colleagues described increased levels of miR-155 in PBMCs from patients with psoriasis compared to controls, which decreased with disease remission [110]. Alatas and colleagues reported a significantly increased expression of miR-155 in patients with psoriasis compared to healthy controls [105]. MiR-155 plays a role in keratinocyte proliferation and the inhibition of apoptosis [111].

The meta-analysis conducted by Zhu and Leung concluded that miR-155 is not a suitable biomarker for T2DM, as 7 studies reported upregulation and 11 studies reported downregulation [71].

In RA, miR-155 expression was increased in patients and correlated with serum and plasma levels of TNF-α and IL-1β. Abdul-Maksoud and colleagues reported increased serum levels of miR-155 that correlated with TNF-α and IL-1β levels in patients with RA [87]. Elmesmari and colleagues also reported increased levels of miR-155 in mononuclear cells from patients with RA compared to healthy controls on a smaller sample [112].

The increased expression of miR-155 in atherosclerosis, psoriasis, and RA and its correlation with TNF-α and IL-1β make it an interesting target for future research and a potential therapeutic target. Studies with more participants are needed to assess the role of miR-155 in vivo.

### 5.6. MiR-145

MiR-145-5p is involved in a variety of human diseases, including cancers, asthma, and rheumatoid arthritis [113]. It is also involved in the phenotype switching of VSMCs in atherosclerosis [114]. MiR-145 is involved in the development of psoriasis and RA via regulating the Wnt/β-catenin pathway [115]. Yuan and colleagues reported that it has an anti-inflammatory function by suppressing the production of IL-1β, TNF-α, and IL-6 after ischemic injury [116].

Studies showed competing evidence for the expression profile of miR-145-5p in atherosclerosis. Minin and colleagues reported increased expression of miR-145-5p in the serum of hypertensive patients with carotid plaques compared to hypertensive patients without carotid plaques [117]. Li and colleagues reported a slight increase in plasma miR-145 levels in patients with atherosclerosis, but this was not statistically significant [108]. However, several studies reported lower miR-145 levels in patients with atherosclerosis. A study by Zhang and colleagues reported a downregulation in miR-145-5p in patients with coronary stenosis [118]. Lv and colleagues also reported decreased plasma levels of miR-145 in patients with an increased brachial–ankle PWV, which is the method used for the early diagnosis of atherosclerosis in China [119]. Meng et al. reported decreased plasma levels of miR-145 in patients with ACS compared to controls and a negative correlation with coronary stenosis [92]. The studies investigated different phenotypes of atherosclerosis which could possibly explain the difference in results obtained. However, only the study by Minin et al. reported an increased expression of miR-145 in cases, and their sample size was smaller than the studies by Meng et al., Lv et al., and Zhang et al. [117,118,119]. Based on the evidence available, we can conclude that miR-145 is downregulated in atherosclerosis, but further research is needed.

Two miR-145 polymorphisms (rs353291 and rs41291957) are also associated with atherosclerosis, which further signifies its role in atherosclerosis [92,120].

The expression of miR-145-5p in the serum of patients with psoriasis was lower compared to healthy controls [115]. Wang and Cao also claimed that the upregulation of miR-145-5p would inhibit the progression of psoriasis [115].

In adipose tissue of patients with T2DM, miR-145-5p is significantly upregulated and was recognized as a potential biomarker for T2DM in the meta-analysis by Zhu and Leung [71].

In RA, miR-145-5p was overexpressed in fibroblast-like synoviocytes and in the plasma of patients with RA [95,121]. Hong et al. reported an increased expression of miR-145 in fibroblast-like synoviocytes (FLS) from patients with RA compared to patients with osteoarthritis [121]. The increased expression of miR-145 promotes the expression of matrix metalloproteinases in FLS in patients with RA [122].

It is difficult to assess the role of miR-145 in the studied diseases; therefore, further studies are needed. The studies conducted in patients with psoriasis, T2DM, and RA were few and relatively small. Because of its reported anti-inflammatory function increasing the expression of miR-145 could be a viable therapeutic option for acute ischemic complications of atherosclerosis as well as ameliorating systemic inflammation.

### 5.7. MiR-200

The miR-200 family (miR-200a, miR-200b, miR-200c, miR-141, and miR-429) is involved in apoptosis, senescence, inflammation, and endothelial dysfunction associated with atherosclerosis [123]. The expression of all members of the miR-200 family is induced by ROS. Members of the MiR-200 family (but especially miR-200c) promote inflammation by inducing ROS, MMP-1, MMP-9, IL-6, and COX-2, and reduce the antioxidant capacity of the cell by decreasing the transcription of manganese superoxide dismutase [123,124].

Plasma levels of miR-200c were significantly increased in patients with carotid plaques and correlated with plaque instability and pro-inflammatory molecules such as MMP-1, MMP-9, IL-6, and COX-2 [124].

In another study, Magenta and colleagues found an upregulation in miR-200c in psoriatic skin lesions and a significant correlation between miR-200c and the severity of psoriasis (PASI score) [123]. In patients with psoriasis, miR-200c also correlates with diastolic dysfunction and cardiac hypertrophy [123]. Circulating levels of miR-200a correlate with arterial stiffness and cardiac hypertrophy in psoriasis patients [123]. Wang and colleagues found a correlation between miR-200a in CD4+ T cells and PASI score, levels of IL-17, and levels of IL-23 [125].

Kujawa and colleagues reported increased levels of miR-200a/b/c in aortic endothelial cells from patients with T2DM compared to healthy controls. They also reported increased endothelial permeability, possibly due to increased levels of the miR-200 family [126]. Ofori and colleagues reported increased levels of miR-200c in pancreatic islets from patients with T2DM compared to controls [127]. The meta-analysis conducted by Zhu and Leung concluded that miR-200 is significantly downregulated in the plasma of patients with T2DM [71]. The meta-analysis included far more participants than the other two studies (242 vs. 10 and 36, respectively). MicroRNA expression is also tissue-specific and this could also explain the difference in the results of the studies. Lo and colleagues also reported decreased miR-200 family levels in aortic endothelial cells stimulated with high glucose levels [128].

No significant difference in the plasma levels of miR-200b-5p and miR-200c-3p was found in patients with RA compared to healthy controls [129].

Further studies are needed to determine the role of miR-200 in atherosclerosis and RA, as only two studies with relatively small sample sizes were found at the time of writing this review. According to the reviewed studies, it is dysregulated in different directions in T2DM and psoriasis. More research has to be performed pertaining to the role of miR-200 in atherosclerosis, psoriasis, RA, and T2DM, but it has the potential to be a therapeutic target because of its involvement in inflammation and oxidative stress.

### 5.8. MiR-133

MiR-133 has three genomic sites in human DNA (on chromosomes 6, 18, and 20) and, together with miR-1, plays a crucial role in heart development [130]. MiR-133a upregulates IL-1β and TNF-α and is associated with increased ROS production in mice [131].

Studies have shown that miR-133 has a crucial regulatory role in important atherosclerotic processes: VSMC differentiation, angiogenesis, inflammation, and apoptosis [132,133,134,135].

Widera et al. reported the increased expression of miR-133 in the plasma of patients with ACS [136]. Patients who had suffered an NSTEMI/STEMI had significantly higher values in comparison to patients with unstable angina pectoris. Their study included 444 participants, but they did not include healthy controls, which is a limitation of their study. They concluded that miR-133 levels were significantly associated with the risk of death [136]. Wang et al. also reported increased miR-133 levels in patients with AMI in comparison with healthy controls and patients with coronary heart disease without AMI, but their sample size was smaller. They also reported a correlation between miR-133 levels and the degree of coronary stenosis [130,137]. The levels of miR-133a and miR-133b negatively correlate with cardio–ankle vascular index in patients with metabolic syndrome [135].

In psoriasis, miR-133 appears to be downregulated. Chicarro and colleagues reported lower miR-133a levels in psoriasis lesions compared to skin samples from healthy controls [138]. The levels of miR-133a-3p increased to those of healthy skin after three months of treatment [138].

De Gonzalo-Calvo et al. and Ghasemi et al. reported significantly increased serum levels of miR-133 in patients with T2DM compared to healthy controls [139,140]. Al-Muhtaresh reported increased miR-133 levels in the whole blood of patients with T2DM and CAD compared to patients with T2DM without CAD [141]. On the other hand, the meta-analysis by Zhu and Leung concluded that miR-133a-3p is significantly downregulated in patients with T2DM [71].

No studies were found that investigated the role of miR-133 in human patients with RA.

The studies reviewed identify miR-133a as an important marker of coronary stenosis in otherwise healthy patients and patients with T2DM. Further studies with larger sample sizes are needed to validate these findings. The only study that investigated the association between miR-133 and psoriasis was related to psoriatic skin lesions. We found no studies examining circulating levels of miR-133 in psoriasis, making comparison with other conditions difficult.

### 5.9. MiR-135

MiR-135 is involved in the regulation of endothelial cell proliferation and apoptosis, inflammation, and angiogenesis [142,143,144]. MiR-135b is upregulated by ox-LDL and is associated with increased levels of TNF-α, IL-1β, IL-6, and IL-8. It also regulates the expression of I-CAM, V-CAM, and E-selectins [142]. Xu and colleagues detected an increased expression of miR-135b-5p in the serum of patients with CAD compared to healthy controls, noting that miR-135b-5p may be involved in the migration and proliferation of VSMCs and endothelial cells [145]. However, Li and colleagues reported a downregulation of miR-135a-5p in patients with coronary atherosclerosis compared to healthy adults [146]. They claimed that it has an atheroprotective function by suppressing the migration and proliferation of VSMCs in atherosclerosis [146].

Joyce and colleagues reported an upregulation of miR-135b in psoriatic skin lesions compared to healthy skin samples [147]. Chicarro and colleagues confirmed these findings and also discovered that miR-135b levels decreased after treatment and correlated with the PASI score [138].

Sarookhani and colleagues reported an increased expression of miR-135a in the plasma of patients with T2DM and prediabetic patients compared to healthy controls [148]. Monfared et al. reported similar results and also found a correlation between serum miR-135a levels and HbA1c levels in prediabetic patients [149]. In another study, Monfared and colleagues reported that miR-135a in saliva could be used as a biomarker for T2DM [150].

Liu and colleagues reported a significant downregulation of miR-135a-5p in synovial tissue from patients with RA compared to healthy controls [151].

To better understand the role of miR-135 in atherosclerosis, psoriasis, T2DM, and RA, further studies with larger sample sizes are needed. All studies examined had relatively small samples and reported conflicting results.

### 5.10. MiR-221

TNF-α induces the expression of miR-221. The function of miR-221-3p in the intima of atherosclerotic vessels is the repression of peroxisome proliferator-activated receptor-γ coactivator 1 α (PGC-1 α) [152]. The post-transcriptional repression of PGC-1 α leads to the accumulation of ROS in endothelial cells and induces apoptosis. MiR-221 is highly expressed in VSMCs and endothelial cells, where it regulates proliferation and apoptosis [153].

Under atherosclerotic conditions, it downregulates eNOS, inhibits vascular repair mechanisms, and promotes VSMC calcification [149]. Minami and colleagues reported elevated levels of miR-221 in the serum of patients with coronary atherosclerosis [154].

MiR-221 is also upregulated in psoriasis and is strongly associated with psoriatic arthritis [155,156,157]. Zibert and colleagues reported an increased expression of miR-221 in psoriatic lesions compared to the skin of healthy controls [155]. Meng et al. and Wade et al. reported increased serum levels of miR-221 in patients with psoriasis and psoriatic arthritis compared to healthy controls [156,157].

MiR-221-3p was significantly upregulated in the serum of patients with T2DM and was reported by Zhu and Leung to be the second most important biomarker for T2DM [71].

The expression of miR-221 is increased in PBMCs and in the serum of patients with RA compared to healthy controls [158,159]. However, Cieśla and colleagues found no significant difference in the plasma levels of miR-221 between patients with RA and healthy controls [160].

MiR-221 is upregulated in all diseases that were the subject of this study. However, only the meta-analysis by Zhu and Leung had a relatively large sample, whereas the other studies included 50 or fewer participants per group. MiR-221 plays an important role in maintaining inflammation. It is induced by one of the major inflammatory cytokines, TNF-α, and causes the accumulation of ROS in cells. Studies in mice have shown that the suppression of miR-221 protects against the development of atherosclerosis [161]. MiR-221 has the potential to be a useful clinical biomarker in the future, but more research is needed to confirm its viability as a therapeutic target.

### 5.11. MiR-424

MiR-424 appears to have an anti-inflammatory function in the human body [162,163]. Its main target is apolipoprotein C3. The suppression of APOC3 inhibits NF-κB and its signaling pathway [164]. It regulates the function of VSMCs in atherosclerosis by regulating the NF-κB signaling pathway [162].

The levels of miR-424-5p are downregulated in the peripheral blood of patients with coronary atherosclerosis [162]. Mir-424-5p is also involved in the accumulation of lipids in foam cells [164].

MiR-424 is involved in the regulation of keratinocyte proliferation [153]. Ichihara et al. reported reduced levels of miR-424 in skin samples and the serum of patients with psoriasis compared to healthy controls [163]. Alatas et al. also reported reduced miR-424 levels in patients with psoriasis compared to healthy controls [107]. MicroRNA-424 levels were significantly higher in the hair shafts of psoriasis patients compared to healthy controls [165].

At the time of writing, there were no published articles discussing miR-424 in patients with T2DM.

Wang and colleagues reported higher levels of miR-424 in synovial samples from patients with RA compared to patients with osteoarthritis [166].

Since miR-424 suppresses the NF-κB signaling pathway, it has the potential to be a therapeutic target in the future. The enhanced suppression of NF-κB by miR-424 could alleviate inflammation and be useful in the treatment of atherosclerosis, psoriasis, T2DM, RA, and other diseases.

### 5.12. Let-7

Let-7 was the second microRNA to be discovered, hence the atypical name. The let-7 family has a tumor-suppressive function and is highly expressed in endothelial cells and VSMCs [123]. They stimulate eNOS, prevent the activation of the NF-κB pathway, and inhibit the apoptosis of endothelial cells [167,168].

In the study by Yu et al. [169], let-7b-5p was downregulated in the serum of patients with coronary atherosclerosis compared to healthy controls. Long and colleagues also report decreased levels of let-7b in patients with large-vessel atherosclerosis who have suffered an ischemic stroke compared to healthy controls and increased levels in patients who have suffered a stroke of another etiology (small-vessel atherosclerosis, cardioembolism, or undetermined etiology) [170]. Huang and colleagues reported a correlation between let-7 serum levels and CIMT in patients with hypertension [171]. An increased expression of let-7c in hypertensive patients with atherosclerosis compared to patients without atherosclerosis was also reported by Minin et al. [117].

Let-7a inhibits INF-γ secretion and the proliferation of T cells in psoriasis [172]. Hu and colleagues discovered a decreased expression of let-7a in patients with psoriasis compared to healthy controls [172]. Pasquali and colleagues also reported a decreased expression of let-7b-5p in patients with psoriasis compared to healthy controls [173]. However, Alatas et al. reported an upregulation of let-7c-5p and let-7d-5p in patients with psoriasis [107].

In patients with CAD and T2DM, plasma levels of let-7b were found to be associated with the regression of T2DM after dietary intervention [174]. In human atherosclerotic plaque tissue, let-7b levels were significantly lower in tissue samples from diabetics compared to non-diabetics [175]. However, Aljaibeji and colleagues reported increased let-7b-5p levels in the serum of patients with T2DM compared to healthy controls [176]. Zhu and Leung found that let-7-f and let-7-i were significantly downregulated in the serum of patients with T2DM in whole blood and serum, respectively [71].

Interestingly, plasma levels of let-7a were significantly increased in patients with RA compared to healthy controls [177]. Ormseth and colleagues included let-7c-5p in a panel of plasma microRNAs that predicted coronary artery calcification in patients with RA. However, no significant association was found between let-7c-5p alone and coronary calcification [178].

We summarize the main findings of the studies that investigated the role of microRNAs in atherosclerosis, psoriasis, T2DM, and RA in Table 1.

MiR-146a, miR-200, and miR-223 were downregulated in patients with T2DM but upregulated in atherosclerosis, psoriasis, and RA. One possible explanation is that they are downregulated by hyperglycemia. Plasma miR-200 levels are decreased by hyperglycemia, but there is no evidence of the effect of hyperglycemia on miR-146a and miR-223, suggesting that further research is needed to test this hypothesis [128].

In the studies examined, miR-143 was upregulated in psoriasis, T2DM, and RA but downregulated in patients with ACS and carotid stenosis. One possible explanation for these results is that impaired blood flow is a stronger mediator of miR-143 expression than inflammation, which occurs in T2DM, RA, and psoriasis.

## 6. Long Non-Coding RNA in Atherosclerosis, Psoriasis, T2DM, and RA

### 6.1. LncRNA-H19

One of the lncRNAs that has been associated with atherosclerosis is lncRNA-H19, which is highly evolutionarily conserved and regulates lipid metabolism, cell proliferation, apoptosis, inflammation, and angiogenesis [179]. It has been shown that LncRNA-H19 promotes the expression of acid phosphatase 5 (ACP5) [180]. Ox-LDL promotes the expression of lncRNA-H19, which promotes endothelial inflammation induced by Ox-LDL [181]. LncRNA-H19 promotes the secretion of TNF-α, IL-1β, and IL-6 and increases ROS production and the expression of ICAM1, VCAM1, and selectins. LncRNA-H19 regulates the NF-κB and MAPK signaling pathways and acts as a molecular sponge for let7 [181].

In patients who had suffered an ischemic stroke and had large artery atherosclerosis, the levels of both lncRNA-H19 and ACP5 were significantly increased compared to patients who had suffered an ischemic stroke and had a different stroke etiology [180]. In another study, lncRNA H19 polymorphism rs217727 was found to be associated with the risk of small vessel ischemic stroke in the Chinese Han population [182]. Bitarafan and colleagues also found elevated levels of lncRNA-H19 in patients with CAD compared to healthy controls, but the difference between the groups was not statistically significant [183]. Several other studies also reported elevated levels of lncRNA-H19 in the serum of patients with atherosclerosis [181,184,185]. However, they did not specify the atherosclerotic phenotype of the participants involved in their studies.

Through its interaction with miR-130b-3p, lncRNA-H19 is also important for the differentiation of keratinocytes and inhibits apoptosis [186]. However, in psoriasis patients, studies have found that lncRNA-H19 is underexpressed in psoriasis lesions compared to healthy skin controls [186,187].

Fawzy and colleagues and Cheng and colleagues reported elevated levels of lncRNA-H19 in the plasma of patients with T2DM compared to healthy controls [188,189]. Similar results were obtained by Tello-Flores and colleagues. They reported increased levels of lncRNA-H19 in the serum of patients with T2DM and poor glycemic control compared to healthy controls [190]. On the other hand, Alfaifi and colleagues reported decreased serum levels of lncRNA-H19 in patients with T2DM compared to healthy controls [191]. Similar results were reported by Alrefai and colleagues, who detected decreased plasma levels of lncRNA-H19 in patients with T2DM [192].

Mahmoudi and colleagues reported significantly higher lncRNA-H19 levels in patients with RA compared to healthy controls [193]. They also reported a significant correlation between lncRNA-H19 and the severity of RA.

The present data show that lncRNA-H19 is significantly elevated in atherosclerosis, but its role in T2DM is inconclusive. Further studies with larger samples are needed to clarify its role in psoriasis, RA, and T2D. Since lncRNA-H19 promotes inflammatory cytokine secretion and ROS production, it could be a viable therapeutic target.

### 6.2. lncRNA-MEG3

LncRNA-MEG3 is upregulated by TNF-α in adipocytes, where it promotes inflammation, and it is downregulated in keratinocytes, where it has an anti-inflammatory function [194,195]. It is involved in endothelial dysfunction, where it enhances inflammation induced by the NLRP3 inflammasome [196]. LncRNA-MEG3 is another lncRNA that appears to be involved in both atherosclerosis and psoriasis, as well as tumor suppression through the accumulation of p53 [197,198,199]. It regulates the proliferation and apoptosis of endothelial cells and vascular smooth muscle cells through miR-26a, miR-21, and miR-223 interactions [199,200]. LncRNA-MEG3 also regulates angiogenesis through the modulation of the VEGF signaling pathway and miR-9 interactions [187]. Studies by Bai et al. and Wu et al. showed a decreased expression of lncRNA-MEG3 in atherosclerotic arteries in patients with CAD [199,200].

The expression of lncRNA-MEG3 was downregulated in skin samples from psoriatic patients compared to healthy skin samples [198]. Jia and colleagues suggested that lncRNA-MEG3 has a direct binding site for miR-21 and, thus, regulates apoptosis and proliferation in psoriatic keratinocytes [198].

Chang and colleagues reported a significant upregulation of lncRNA-MEG3 in PBMCs in patients with T2DM and vascular complications compared to patients with T2DM without vascular complications and controls [201].

In patients with RA, levels of lncRNA-MEG3 were elevated in plasma, PBMCs, and synovial fluid compared to healthy controls [202]. However, Wahba and colleagues reported a downregulation of lncRNA-MEG3 in the serum of patients with RA compared to healthy controls. They also reported that the rs941576 polymorphism of lncRNA-MEG3 is associated with increased severity of RA in Egyptian patients [203].

The studies that investigated the role of lncRNA-MEG3 in atherosclerosis and psoriasis did so on tissue samples (atherosclerotic or psoriatic plaques), whereas the studies that investigated its role in T2DM and RA used PBMCs or serum of the participants. This and its tissue-specific role in inflammation makes comparison between the studies more difficult.

### 6.3. lncRNA-UCA1

LncRNA-UCA1 downregulates NF-κB [204]. The role of lncRNA-UCA1 in psoriasis is not clearly understood. Ma and colleagues reported the decreased expression of lncRNA-UCA1 in the lesional skin of patients with psoriasis in comparison to the non-lesional skin of the same patients [205]. Tian and colleagues reported that lncRNA-UCA1 is upregulated in vascular smooth muscle cells (VSMCs) treated with oxidized LDL and directly affects VSMC proliferation in atherosclerosis by modulating miR-26a expression [206].

In T2DM patients, lncRNA-UCA1 was downregulated in serum, serum exosomes, and VSMCs compared to healthy controls [207]. It has been reported to promote VSMC proliferation via miR-582-5p under hyperglycemic conditions.

LncRNA-UCA1 was underexpressed in fibroblast-like synoviocytes from patients with RA compared to healthy controls [208].

### 6.4. LncRNA-XIST

The LncRNA-X-inactive specific transcript (lncRNA-XIST) is important for the inactivation of the X chromosome in female mammals in the placenta [209]. It is also involved in inflammation and carcinogenesis in various types of cancer. Higher levels of lncRNA-XIST have been associated with poorer prognosis in solid tumors [210,211].

Wang and colleagues reported higher lncRNA-XIST levels in patients with psoriasis compared to healthy controls and a positive correlation between lncRNA-XIST levels and the PASI score, as well as TNF-α, IL-17, and IL-22 levels [211]. To date, only the role of lncRNA-XIST in the transformation of atherosclerotic VSMCs is known. It promotes their migration and proliferation and reduces apoptosis [212,213,214].

Patients with T2DM have been reported to have a decreased expression of lncRNA-XIST compared to healthy controls [215,216]. Interestingly, lncRNA-XIST was significantly upregulated in patients with CAD who also had T2DM compared to patients with CAD but without T2DM [217].

Liu and colleagues reported an increased expression of lncRNA-XIST in synovial tissues of patients with RA compared to healthy controls [218].

We summarize the main findings of the studies that have investigated the role of lncRNAs in atherosclerosis, psoriasis, T2DM, and RA in Table 2. This area of research is underdeveloped, but we expect more studies to be published in the future. The main limitation of the studies examined is the small sample sizes. LncRNA-MEG3 is upregulated in T2DM but downregulated in atherosclerosis and psoriasis (the two studies examining lncRNA-MEG3 in RA reported conflicting results). This could be due to the fact that hyperglycemia upregulates lncRNA-MEG3; however, experimental studies are needed to evaluate this hypothesis.

## 7. The Role of Extracellular Vesicles in Atherosclerosis, Psoriasis, T2DM, and RA

Microvesicles not only act as biomarkers for disease but also contribute to inflammation and promote the pathogenesis of atherosclerosis [219]. In vitro studies have shown that MVs produced under pathological conditions promote endothelial dysfunction [220,221]. Patients with psoriasis have higher concentrations of circulating EVs of endothelial origin of all sizes and higher concentrations of platelet-derived MVs of less than 500 nm in size in their blood compared to healthy controls [222]. Zhang and colleagues reported increased plasma concentrations of platelet-derived MVs in patients with T2DM compared to obese and healthy subjects [223].

Increased levels of platelet-derived CD41+ MVs were found in patients with peripheral artery disease compared to healthy individuals [224]. In a hospital-based cross-sectional study conducted on 40 psoriasis patients and 12 healthy participants, a significant correlation was found between platelet-derived MVs and IL-12 and IL-17. CD41+ MVs were also significantly more common in psoriasis patients [225]. Tamagawa-Mineoka and colleagues discovered that the amount of platelet-derived MVs strongly correlated with the severity of psoriasis; however, their study was only conducted in 21 psoriasis patients and 22 healthy controls [226]. Ho and colleagues found increased levels of CD41+ and CD31+ platelet MVs compared to healthy controls [227]. Platelets with such surface markers have been associated with unstable atherosclerotic plaques [26].

Endothelial and platelet MV levels are decreased in psoriasis patients treated with anti-TNF-α medications, which may reduce the risk of cardiovascular complications [228,229].

Tan and colleagues reported increased levels of platelet MVs in patients with T2DM and clinically apparent atherosclerosis compared to healthy controls and T2DM patients without clinically apparent atherosclerosis [230].

MVs derived from the platelet-poor plasma of patients with RA promoted the production of IL-6 and IL-8 and the expression of CD54+ in endothelial cells, as well as the adhesion of monocytes to the endothelium [231]. Michael and colleagues reported significantly higher levels of platelet- and leukocyte-derived MVs in patients with RA compared to patients with osteoarthritis and healthy controls. They found no correlation between MV levels and disease activity [232].

The concentrations of endothelial cell-derived EVs are elevated in both psoriasis and atherosclerosis and correlate with disease severity [27,233]. EVs produced by endothelial cells can induce the expression of adhesion molecules and selectins involved in leukocyte diapedesis [234]. Endothelial apoptotic bodies promote atherosclerosis via their miR-126 cargo; however, they may also have anti-inflammatory and cytoprotective functions [26]. MiR-126 may inhibit inflammation by mediating the production of VCAM-1, thus reducing inflammation.

The exact role of miR-126 in psoriasis is still debatable. Murzina and colleagues discovered that levels of miR-126 correlate with disease severity and response to treatment in children with psoriasis [235]. Feng and colleagues also discovered a positive correlation between miR-126 and the severity of psoriasis and inflammation [236]. However, Pelosi and her research team and Duan and colleagues reported a negative correlation between miR-126 levels and the risk of developing psoriasis and the severity of the disease [237,238].

Levels of CD105+ endothelial EVs are significantly increased in psoriasis patients compared to healthy controls [239]. Chironi and colleagues reported a correlation between CD105+ EVs and CIMT [240]. Kandiyil and colleagues discovered an association between CD105+ EVs and either stroke severity or clinical outcome [241]. Several studies have also reported an increased expression of CD105+ in unstable atherosclerotic plaques [242,243,244].

Marei and colleagues reported significantly increased levels of CD42− CD31+ endothelial EVs in patients with T2DM and ACS compared to healthy controls [245]. Rodríguez-Carrio and colleagues reported increased levels of total EVs in patients with RA compared to healthy controls and a correlation with cardiovascular risk factors [246].

Smooth muscle cells, endothelial cells, and macrophages are all affected by senescence in the process of atherosclerosis. Even though these cells are in the last phase of their life cycle, they still secrete EVs and pro-inflammatory cytokines. EVs carrying miR-21 induce senescence in endothelial cells, implying that they may also be involved in the progression of atherosclerosis [26]. This is possibly another overlap between the pathological pathways of psoriasis and atherosclerosis, as there is evidence that miR-21 is upregulated in psoriasis [23]. In atherosclerotic conditions, platelet-derived EVs stimulate the activity of IL-1, IL-6, and IL-8, all of which are important mediators in psoriasis [247,248]. EVs secreted by macrophages, neutrophils, mesenchymal stem cells, keratinocytes, and adipocytes have also been associated with inflammation [235].

Figure 2 schematically represents the role of extracellular vesicles in the pathological processes involved in atherosclerosis and psoriasis.

## 8. Conclusions

It has been well established that patients who suffer from inflammatory diseases, such as T2DM, RA, and psoriasis, have an increased risk of developing atherosclerosis. According to the 2021 ESC Guidelines on cardiovascular disease prevention, T2DM, RA, and psoriasis all increase the risk of CVD [12]. RA increases the risk of CVD, calculated using SCORE2/SCORE2-OP, by 50%. The increased risk is independent of the factors used to calculate the risk (age, sex, smoking status, blood pressure, serum non-HDL cholesterol, population-based risk) [12]. T2DM increases the CVD risk by up to twofold (when diagnosed in early adulthood) and is associated with a high or very high risk for CVD in the next 10 years in the majority of T2DM patients [12,249]. A large cohort study performed in the UK reported an increased risk of major adverse cardiovascular events in patients with psoriasis (those taking disease-modifying antirheumatic drugs (DMARDs) and those not) even after adjusting for traditional risk factors (age, sex, hypertension, diabetes, hyperlipidemia, and smoking status) [250].

Systemic chronic inflammation induces the formation of focal atherosclerotic plaques at predilection sites (i.e., bifurcations of the arteries). Shear stress exerted on the endothelium at these sites alters the expression of microRNAs and endothelium-derived microvesicles [251]. Hyperglycemia also alters the regulation of different microRNAs (e.g., miR-27a-3p, miR-29, miR-92a, miR-200) [128,252,253]. MiR-27a-3p was shown to be responsible for maintaining the hyperglycemic metabolic memory even in normoglycemic conditions [253]. MicroRNAs and lncRNAs regulate and are regulated by inflammatory cytokines through complex feedback loops. Targeting inflammatory cytokines to treat atherosclerotic disease has proven effective; however, high costs of treatment and adverse side effects are the main issues. Specific miRNA therapeutics are already in different stages of development for various diseases (hepatitis C, various cancers, Alport syndrome) [254]. As technology develops and becomes more accessible, we can expect miRNA-based medication development for the treatment of atherosclerosis and its complications. In order for such treatments to be established, they would have to be safe, more efficient, and cost-effective than the already established ones, so they will more likely be developed for acute complications (myocardial infarction and stroke) first. In the future, microRNA-based therapy could be used as an add-on to conventional therapy in patients who could not meet the clinical objectives using only conventional therapy.

In this paper, we reviewed the studies that investigated the associations between selected miRs and lncRNAs in atherosclerosis, psoriasis, T2DM, and RA. The interest in researching microRNAs, long non-coding RNAs, and extracellular vesicles has increased recently because of the advancements in technology and the potential for novel treatment options. Information about the molecular profiles of EVs and their specific function in pathological pathways is still sparse. Advancements in the isolation of EVs and subsequent analyses are needed in order to move this field one step closer to clinical implementation. Further research with a larger number of participants is needed since most of the studies reviewed had a relatively small sample size.

Even though inflammation and immune response affect the development and progression of atherosclerotic disorders (CAD, carotid disease, peripheral artery disease) as well as other inflammatory disorders (psoriasis, T2DM, and RA), an important synergistic effect of inflammation/inflammatory markers (miRs, lncRNAs, microvesicles, etc.) on the development and progression of atherosclerosis might be expected. Measures to decrease inflammatory response are needed, and knowledge about the role of different inflammatory markers is mandatory.

## Figures and Tables

**Figure 1 biomedicines-12-01322-f001:**
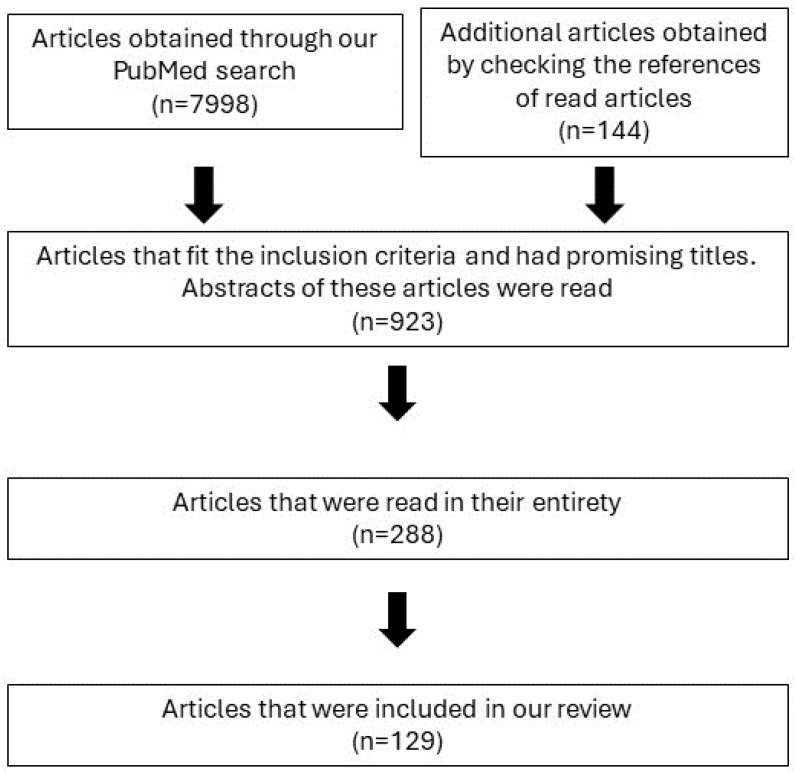
Schematic representation of our article selection process.

**Figure 2 biomedicines-12-01322-f002:**
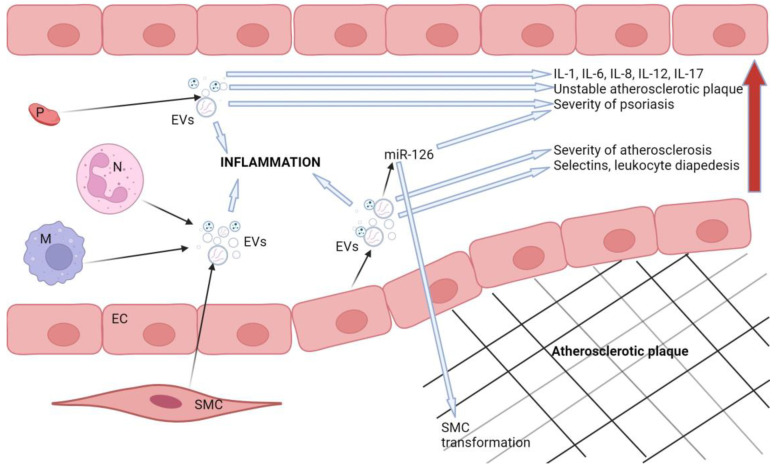
Schematic representation of the role of extracellular vesicles in the pathogenesis of atherosclerosis and psoriasis. The red arrow represents the propagation of harmful effects caused by extracellular vesicles. Legend: SMC: smooth muscle cell, M: macrophages, N: neutrophils, EVs: extracellular vesicles, P: platelet, EC: endothelial cell. Created with BioRender.com.

**Table 1 biomedicines-12-01322-t001:** Summary of the clinical studies that investigated the roles of microRNAs and were reviewed in this article.

Type of MicroRNA	Atherosclerosis			Psoriasis			Type 2 Diabetes Mellitus			Rheumatoid Arthritis		
	Authors of the Study	No. of Patients/Samples (Cases/Controls)	Main Findings	Authors of the Study	No. of Patients (Cases/Controls)	Main Findings	Authors of the Study	No. of Patients (Cases/Controls)	Main Findings	Authors of the Study	No. of Patients (Cases/Controls)	Main Findings
miR-146a	Huang et al. [57]	180/90	↑	Leal et al. [61]	99/78	↑	Alipoor et al. [70]	344/316	↓	Bae et al. [74]	683/477	↑
	Raitoharju et al. [63]	30/20	↑	Ele-Refaei et al. [62]	40/10	↑	Zhu and Leung [71]	Adipose 24	↑			
	Zhelankin et al. [64]	50/30	↑	Vaher et al. [60]	26/26	↑	Zhu and Leung [71]	PBMC 140	↓			
	Guo et al. 65]	42/58	↑									
	Gao et al. [66]	56/56	↑									
miR-210	Signorelli et al. [79]	27/27	↑	Zhao et al. [81]	18/18	↑	Li et al. [82]	32/32	↑	Abdul-Maksoud et al. [87]	100/100	↓
	Raitoharju et al. [63]	30/20	↑	Wu et al. [78]	30/30	↑	Amr et al. [84]	100/20	↑	Huang et al. [88]	38/45	x
							Zhou et al. [85]	10/10	↓			
							Zhang et al. [86]	32/20	↓			
							Zhu and Leung [71]	540	↑			
miR-143	Meng et al. [92]	279/65	↓	Løvendorf et al. [93]	55/33	↑	Zhu and Leung [71]	112	↑	Yue et al. [95]	117/6	↑
				Zheng et al. [94]	194/175	↓						
miR-223	Singh et al. [100]	250/250	↑	Løvendorf et al. [93]	55/33	↑	Parrizas et al. [99]	1184/838	↓	Ormseth et al. [106]	168/91	↑
	Guo et al. [101]	300/100	↑	Pivarcsi et al. [104]	43/22	x	Zhu and Leung [71]	Plasma 309	↓			
	Zhu et al. [102]	52/25	↓	Alatas et al. [105]	52/54	↓						
miR-155	Fichtlscherer et al. [107]	31/14	↓	García-Rodríguez et al. [110]	11/11	↑	Zhu and Leung [71]	Whole blood 120	↑	Abdul-Maksoud et al. [87]	100/100	↑
	Li et al. [108]	70/55	↑	Alatas et al. [105]	52/54	↑	Zhu and Leung [71]	PBMC 80	↓	Elmesmari et al. [112]	24/22	↑
	Wang et al. [109]	3/x										
miR-145	Meng et al. [92]	279/65	↓	Wang et al. [115]	45/40	↓	Zhu and Leung [71]	24	↑	Yue et al. [95]	117/6	↑
	Minin et al. [117]	105/72	↑							Hong et al. [121]	5/5	↑
	Lv et al. [119]	328/374	↓									
	Li et al. [108]	70/55	x									
	Zhang et al. [118]	207/66	↓									
miR-200	Magenta et al. [124]	24/19	↑	Magenta et al. [124]	29/29	↑	Kujawa et al. [126]	5/5	↑	Balzano et al. [129]	28/20	x
				Wang et al. [125]	189/109	↑	Ofori et al. [127]	9/27	↑			
							Zhu and Leung [71]	Plasma 242	↓			
miR-133	Wang et al. [137]	154/92	↑	Chicharro et al. [138]	44/5	↓	De Gonzalo-Calvo et al. [139]	72/x	↑	-	-	-
	Al-Muhtaresh et al. [141]	30/30	↑				Ghasemi et al. [140]	35/35	↑			
							Al-Muhtaresh et al. [141]	30/30	↑			
							Zhu and Leung [71]	169	↓			
miR-135	Xu et al. [145]	77/45	↑	Chicharro et al. [138]	44/5	↓	Sarookhani et al. [138]	30/30	↑	Liu et al. [151]	3/x	↓
	Li et al. [146]	47/47	↓	Joyce et al. [147]	26/26	↑	Monfared et al. [149]	80/40	↑			
miR-221	Minami et al. [154]	44/22	↑	Zibert et al. [155]	13/13	↑	Zhu and Leung [71]	Serum 793	↑	Abo ElAtta et al. [158]	30/20	↑
				Meng et al. [156]	46/42	↑				Cunningham et al. [159]	50/20	↑
				Wade et al. [157]	31/20	↑				Ciesla et al. [160]	50/24	x
miR-424	Li et al. [162]	75/60	↓	Ichihara et al. [163]	15/15	↓	-	-	-	-	-	-
				Alatas et al. [105]	52/54	↓						
let-7	Long et al. [170]	179/50	↑/↓	Alatas et al. [105]	52/54	↑	Aljaibeji et al. [176]	29/25	↑	Cunningham et al. [159]	50/20	↑
	Huang et al. [171]	60/60	↑	Hu et al. [172]	40/38	↓	Zhu and Leung [71]	Let-7-f Whole blood 89	↓	Tang et al. [177]	76/36	↑
	Minin et al. [117]	105/72	↑				Zhu and Leung [71]	Let-7-i serum 54	↓			
	Yu et al. [169]	30/22	↓									

Legend: ↑: increased expression/upregulation, ↓: decreased expression/downregulation, x: there was no statistically significant difference.

**Table 2 biomedicines-12-01322-t002:** Summary of the clinical studies that investigated the roles of lncRNAs and were reviewed in this article.

Type of LncRNA	Atherosclerosis			Psoriasis			Type 2 Diabetes Mellitus			Rheumatoid Arthritis		
	Authors of the Study	No. of Patients/Samples (Cases/Controls)	Main Findings	Authors of the Study	No. of Patients (Cases/Controls)	Main Findings	Authors of the Study	No. of Patients (Cases/Controls)	Main Findings	Authors of the Study	No. of Patients (Cases/Controls)	Main Findings
H19	Cao et al. [181]	27/20	↑	Gupta et al. [187]	18/16	↓	Fawzy et al. [188]	119/110	↑	Mahmoudi et al. [193]	25/25	↑
	Bitarafan et al. [183]	50/50	↑	He et al. [186]	6/6	↓	Cheng et al. [189]	30/30	↓			
	Han et al. [185]	30/30	↑				Tello-Flores et al. [190]	60/60	↑			
	Huang et al. [180]	80/85	↑				Alfaifi et al. [191]	200/200	↓			
	Pan et al. [184]	42/37	↑				Alrefai et al. [192]	65/65	↓			
MEG3	Bai et al. [199]	40/35	↓	Jia et al. [198]	19/19	↓	Alrefai et al. [192]	65/65	↑	Chatterjee et al. [202]	82/15	↑
	Wu et al. [200]	15/15	↓				Chang et al. [201]	53/62	↑	Wahba et al. [203]	100/100	↓
UCA1	-	-	-	Ma et al. [205]	20/x	↓	Yang et al. [207]	40/40	↓	-	-	-
XIST	Sohrabifar et al. [217]	25/25	x	-	-	-	Wang et al. [215]	76/76	↓	Liu et al. [218]	20/7	↑
							Sohrabifar et al. [217]	25/25	↑			

Legend: ↑: increased expression/upregulation, ↓: decreased expression/downregulation, x: there was no statistically significant difference.

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
