# Peer review of "A Review of MicroRNAs and lncRNAs in Atherosclerosis as Well as Some Major Inflammatory Conditions Affecting Atherosclerosis"

_biomedicines, 2024, doi:10.3390/biomedicines12061322_

Round 1
Reviewer 1 Report
Comments and Suggestions for Authors
I read with great interest the paper “MicroRNAs and lncRNAs in Atherosclerosis: Associations with Major Inflammatory Conditions”.
English is generally sound, only minor revision.
1. The abstract provides a clear overview of the aims and content of the review. It succinctly outlines the associations between microRNAs (miRs), long non-coding RNAs (lncRNAs), and various inflammatory conditions such as psoriasis, type 2 diabetes mellitus (T2DM), and rheumatoid arthritis (RA), in the context of atherosclerosis. It also mentions the role of extracellular vesicles (EVs) in intracellular signal transduction. However, the abstract could be improved by providing more specific details about the findings or conclusions of the review. Additionally, some sentences could be rephrased for clarity and flow. For example, the sentence "Despite the fact that inflammation and immune response affect the development and progression of atherosclerotic disorders...," could be simplified for easier comprehension.
2. The introduction effectively outlines the role of inflammation in atherosclerosis and its association with various inflammatory conditions. However, it could benefit from clearer structuring to improve readability. To strengthen the introduction, consider including more specific details or statistics to support the statements made. For example, providing statistics on the prevalence of these diseases or the impact of inflammation on disease progression could enhance the discussion.
3. The text regarding microRNA mentions contradictory findings in some studies, particularly regarding the expression levels of certain microRNAs in different diseases. It's important to acknowledge conflicting results, but it would be beneficial to discuss potential reasons for these discrepancies, such as differences in study populations, methodologies, or disease subtypes. Moreover, while the text provides a summary of various studies, it lacks critical analysis or interpretation of the findings. For instance, instead of just listing increased or decreased expression levels of microRNAs, discussing the potential implications of these findings on disease pathogenesis or treatment could add depth to the review. Furthermore, the text mentions that some studies had small sample sizes, especially in diseases like psoriasis, T2DM, and RA. It's essential to consider the quality of evidence provided by studies with small sample sizes, as they may have limited generalizability or statistical power.
4. While the text discusses the potential therapeutic implications of targeting inflammatory markers such as miRNAs and lncRNAs, it could further elaborate on the practical implications for clinical practice. For example, how might the modulation of these molecules be incorporated into existing treatment strategies for inflammatory diseases? Please reade paragraph 2.7 of the following review (https://doi.org/10.3390/biomedicines10092274)
5. The text briefly mentions the need for advancements in technology and larger-scale studies to further explore the roles of miRNAs, lncRNAs, and extracellular vesicles in inflammatory diseases. Providing specific recommendations for future research directions, such as the exploration of novel therapeutic targets or the development of diagnostic biomarkers, would be valuable.
Comments on the Quality of English LanguageOnly minor issues
Author Response
English is generally sound, only minor revision.
- The abstract provides a clear overview of the aims and content of the review. It succinctly outlines the associations between microRNAs (miRs), long non-coding RNAs (lncRNAs), and various inflammatory conditions such as psoriasis, type 2 diabetes mellitus (T2DM), and rheumatoid arthritis (RA), in the context of atherosclerosis. It also mentions the role of extracellular vesicles (EVs) in intracellular signal transduction. However, the abstract could be improved by providing more specific details about the findings or conclusions of the review. Additionally, some sentences could be rephrased for clarity and flow. For example, the sentence "Despite the fact that inflammation and immune response affect the development and progression of atherosclerotic disorders...," could be simplified for easier comprehension.
Thank you for this comment. We have modified the abstract accordingly.
- The introduction effectively outlines the role of inflammation in atherosclerosis and its association with various inflammatory conditions. However, it could benefit from clearer structuring to improve readability. To strengthen the introduction, consider including more specific details or statistics to support the statements made. For example, providing statistics on the prevalence of these diseases or the impact of inflammation on disease progression could enhance the discussion.
Thank you for your comment. We have included statistics on the prevalence of the diseases and reformed the structure.
- The text regarding microRNA mentions contradictory findings in some studies, particularly regarding the expression levels of certain microRNAs in different diseases. It's important to acknowledge conflicting results, but it would be beneficial to discuss potential reasons for these discrepancies, such as differences in study populations, methodologies, or disease subtypes. Moreover, while the text provides a summary of various studies, it lacks critical analysis or interpretation of the findings. For instance, instead of just listing increased or decreased expression levels of microRNAs, discussing the potential implications of these findings on disease pathogenesis or treatment could add depth to the review. Furthermore, the text mentions that some studies had small sample sizes, especially in diseases like psoriasis, T2DM, and RA. It's essential to consider the quality of evidence provided by studies with small sample sizes, as they may have limited generalizability or statistical power.
Thank you for your insightful comment, we have changed the article according to your suggestions.
- While the text discusses the potential therapeutic implications of targeting inflammatory markers such as miRNAs and lncRNAs, it could further elaborate on the practical implications for clinical practice. For example, how might the modulation of these molecules be incorporated into existing treatment strategies for inflammatory diseases? Please reade paragraph 2.7 of the following review (https://doi.org/10.3390/biomedicines10092274)
“The endothelial layer of the coronary microvasculature is directly exposed to hemodynamic forces such as arterial pressure, which is frequently elevated in diabetes, and shear stress, both of which may induce vasomotor changes and remodeling [
- The text briefly mentions the need for advancements in technology and larger-scale studies to further explore the roles of miRNAs, lncRNAs, and extracellular vesicles in inflammatory diseases. Providing specific recommendations for future research directions, such as the exploration of novel therapeutic targets or the development of diagnostic biomarkers, would be valuable.
- Thank you for your suggestions, we have expanded the article accordingly.
Reviewer 2 Report
Comments and Suggestions for Authors
The authors aimed to summarize the associations between selected microRNAs (miRs) and long non-coding RNAs (lncRNAs), and atherosclerosis, psoriasis, type 2 diabetes, and rheumatoid arthritis.
They concluded that based on the finding an important synergistic effect of inflammation /inflammatory markers (miRs, lncRNAs, microvesicles, etc) on the development and progression of atherosclerosis might be expected. Measures to decrease inflammatory response are needed, and the knowledge about the role of different inflammatory markers are mandatory.
Comments:
· Introduction: the role and significance of miRs, lncRNAs, microvesicles should be mentioned.
· The Methods section should be completed with a chart flow demonstrating the process of literature review.
· Clinical significance and potential future therapeutic role of miRs, lncRNAs, microvesicles should be highlighted.
· Some abbreviations are not explained, such as WMD, LDL, HDL, VSMC, CAD, PBMC, etc… A list of abbreviation could be useful.
· Author Contributions: must be corrected.
Comments on the Quality of English LanguageMinor editing of English language required.
Author Response
The authors aimed to summarize the associations between selected microRNAs (miRs) and long non-coding RNAs (lncRNAs), and atherosclerosis, psoriasis, type 2 diabetes, and rheumatoid arthritis.
They concluded that based on the finding an important synergistic effect of inflammation /inflammatory markers (miRs, lncRNAs, microvesicles, etc) on the development and progression of atherosclerosis might be expected. Measures to decrease inflammatory response are needed, and the knowledge about the role of different inflammatory markers are mandatory.
Comments:
- Introduction: the role and significance of miRs, lncRNAs, microvesicles should be mentioned.
Thank you for your comment, we have added some basic information regarding miRs, lncRNAs and microvesicles in the introduction.
- The Methods section should be completed with a chart flow demonstrating the process of literature review.
Thank you for this suggestion, we have included a flow chart.
- Some abbreviations are not explained, such as WMD, LDL, HDL, VSMC, CAD, PBMC, etc… A list of abbreviation could be useful.
We have included a list of abbreviations at the end of the article.
- Author Contributions: must be corrected.
Thank you for noticing this, we have corrected the Author contributions.
Clinical significance and potential future therapeutic role of miRs, lncRNAs, microvesicles should be highlighted.
Thank you for your suggestion, we have changed the article accordingly.
Round 2
Reviewer 1 Report
Comments and Suggestions for Authors
Point four has been poorly answered in the text. Please expand it and cite the suggested manuscript.
Comments on the Quality of English Languageminor spell check
Author Response
To the reviewer:
Paper was revised as suggested and we added missing information (marked in yellow).
Moreover, English was checked again